# Within-session test-retest reliability of pressure pain threshold and mechanical temporal summation in healthy subjects

Catherine Mailloux[1]☺, Louis-David Beaulieu[2‡], Timothy H. Wideman[3‡], Hugo Massé-Alarie[1]☺*

**1** Centre Interdisciplinaire de Recherche en Réadaptation et Intégration Sociale (CIRRIS), Université Laval, Quebec, Canada, **2** BioNR Research Lab, Université du Québec à Chicoutimi, Chicoutimi, Canada, **3** School of Physical and Occupational Therapy, McGill University, Montreal, Canada

☺ These authors contributed equally to this work.
‡ These authors contributed also equally to this work.
* hugo.masse-alarie@fmed.ulaval.ca

**Data Availability Statement:** All files are available from the Scholar Portal Dataverse database (https://doi.org/10.5683/SP2/URLJLN, Scholars

## Abstract

### Objective

To determine the absolute and relative intra-rater within-session test-retest reliability of pressure pain threshold (PPT) and mechanical temporal summation of pain (TSP) at the low back and the forearm in healthy participants and to test the influence of the number and sequence of measurements on reliability metrics.

### Methods

In 24 participants, three PPT and TSP measures were assessed at four sites (2 at the low back, 2 at the forearm) in two blocks of measurements separated by 20 minutes. The standard error of measurement, the minimal detectable change (MDC) and the intraclass correlation coefficient (ICC) were investigated for five different sequences of measurements (e.g. measurement 1, 1–2, 1-2-3).

### Results

The MDC for the group (MDC$_{gr}$) for PPT ranged from 28.71 to 50.56 kPa across the sites tested, whereas MDC$_{gr}$ for TSP varied from 0.33 to 0.57 out of 10 (numeric scale). Almost all ICC showed an excellent relative reliability (between 0.80 and 0.97), except when only the first measurement was considered (moderate). Although minimal differences in absolute PPT reliability were present between the different sequences, in general, using only the first measurement increase measurement error. Three TSP measures reduced the measurement error.

### Discussion

We established that two measurements of PPT and three of TSP reduced the measurement error and demonstrated an excellent relative reliability. Our results could be used in future

Portal Dataverse, V1, UNF:6:g
+Lz6E4IvbaXIy5wuxkzPQ==).

**Funding:** This study is financed by a grant from the Quebec Pain Research Network (QPRN; https://qprn.ca/fr), the Réseau Provincial de Recherche en Adaptation-Réadaptation (REPAR; https://repar.ca) and the Canadian Musculoskeletal Rehab Research Network (http://mskrehabnet.com). HMA is supported by salary awards from Fonds de recherche du Québec - Santé (281961 - FRQS; http://www.frqs.gouv.qc.ca). CM received grants from the Canadian Institute of Health Research (CIHR; https://cihr-irsc.gc.ca/e/193.html) and the Ordre professionnel de la physiothérapie du Québec (OPPQ; https://oppq.qc.ca). The funders had no role in study design, data collection and analysis, decision to publish, or preparation of the manuscript.

**Competing interests:** The authors have declared that no competing interests exist.

pain research to confirm the presence of true hypo/hyperalgesia for paradigms such as conditioned pain modulation or exercise-induced hypoalgesia, indicated by a change exceeding the measurement variability.

## 1. Introduction

The experience of pain is highly variable and influenced by biological, psychological and social factors [1]. One essential feature of the experience of pain is the capacity of the nervous system to modulate pain through the interplay of multiple areas and mechanisms [2]. The complexity of pain modulation mechanisms makes it difficult to evaluate. In humans, psychophysical experimental paradigms have been developed in research as proxy of pain modulation. Conditioned pain modulation (CPM) and exercise-induced hypoalgesia (EIH), for instance, are used to approximate the efficacy of pain inhibition [3–5]. Recently, these paradigms have been increasingly used in research to determine if individuals with chronic pain have an altered pain inhibition response. For example, some studies reported an alteration of pain inhibition in individuals with chronic low back pain (CLBP) using CPM [6–9] and EIH [10,11].

CPM refers to the decrease in pain sensitivity after the administration of a painful stimulus on a remote body part (e.g. cold water immersion of the hand [12]). EIH represents the decrease in pain sensitivity that occurs following an isometric, resistance or aerobic exercise [13]. Opioidergic, serotonergic and noradrenergic systems contribute to both CPM [2,14–17] and EIH [18–24]. Experimental protocols developed to assess EIH and CPM take generally part in a within-session design, including pain sensitivity measures collected before and after a conditioning stimulus, and they are determined by the change in pain sensitivity between test and retest. The pressure pain threshold (PPT) is the most used pain sensitivity measure for these paradigms [13,25,26]. PPT is a static measure of pain and would reflect the basal state of pain perception [4,27]. Temporal summation of pain (TSP) has also been used as a pain sensitivity measure to quantify CPM/EIH [28–30] and rather constitutes a dynamic measure of pain sensitivity that refers to the perception of increasing pain in response to stable (continuous or repeated) noxious stimuli [31–33]. To ensure the validity of these paradigms, it is essential that pain sensitivity measures chosen as testing stimulus (PPT and TSP) present small measurement error. This permits to define a minimal level of true change, exceeding the measurement variability, to confirm the presence of hyper- or hypoalgesia following a conditioning stimulus. In addition, reliability needs to be considered in respect of the site tested. Therefore, with a view of studying CLBP population, measuring variability of PPT and TSP at the low back area and at a remote site is essential to provide a general view of pain sensitivity and pain modulation functioning. Some studies documented PPT reliability at the low back [34–37], but the sample size tested in these studies remains limited. For TSP, no study measured the reliability at the lower back area. Moreover, the number of measures needed to reach an acceptable level of test-retest reliability for TSP is not known. This is essential to ensure the applicability and the feasibility of measuring PPT and TSP in research and clinical practice, and to reduce the number of painful stimulus needed to obtain reliable and valid data.

The objectives are to 1) determine the absolute and relative intra-rater within-session test-retest reliability of PPT and mechanical TSP at the low back and at the forearm in healthy participants and 2) test the influence of the number and sequence of measurements for PPT and TSP on reliability metrics. Our study focused on absolute reliability that will provide the minimal level to reach a real change in pain sensitivity, which is essential to interpret the presence/absence of hyper/hypoalgesia using pain modulation paradigms such as CPM and EIH.

## 2. Materials and methods

### 2.1 Participants

Twenty-four healthy subjects (12 females | 12 males; 28.3 ± 11.0 years old) aged between 18 and 65 years old were recruited between December 2018 and June 2019 (see **Table 1** for descriptive statistics). Sampling method was by convenience with emails sent to Laval University community (60,000 individuals comprising students and employees) and by solicitation at the research center. Determination of selection criteria was based on consensus statement by the EUROPAIN and NEUROPAIN consortia for quantitative sensory testing (QST)-based studies to ensure validity of the data [38]. Exclusion criteria were: 1) pain lasting three months or longer, located anywhere in the body, 2) severe health problem (such as cancer, major rheumatoid, cardiac, neurologic or psychiatric disease), 3) low back pain lasting more than 7 days in the last 6 months, 4) consultation with a health professional because of low back pain in the last 6 months, 5) current bilateral wrist or forearm pain and 6) current pregnancy and/or gave birth in the last year. Subjects who were currently taking medication like antidepressants, opioids, neuroleptics, anticonvulsive drugs or steroids were also excluded. This study was approved by local ethic committee (CIUSSS-Capitale Nationale, project #2019–1547) and all participants provided informed written consent prior experimentation. The body mass index was calculated for each participant and the Global Physical Activity Questionnaire (GPAQ) was self-administered to rate the level of physical activity [39].

### 2.2 Study design

All measurements were collected in a single session at the research center by the same rater (CM). The rater was a physical therapist with five years of clinical experience who had undertaken a QST training with experienced researchers.

PPT and TSP were tested in two blocks, lasting approximately 15 minutes each, separated by a pause of 20 minutes (**Fig 1**). The 20 min-pause was established in consideration of the time interval duration between blocks of testing when assessing EIH and CPM (e.g. time interval between the measure of PPT pre to post conditioning stimulus) [26,40–43]. This was done to ensure that PPT and TSP reliability were tested in a design reproducing CPM/EIH protocols.

**Table 1. Descriptive statistics (n = 24).**

| | |
|---|---|
| Age, years, $\bar{X}$ ± SD | 28.3 ± 11.0 |
| Sex (male: female) | 12: 12 |
| Race, % | |
| Caucasian | 92 |
| Asian | 4 |
| Black | 4 |
| BMI | 23.0 ± 3.12[†] |
| GPAQ | 2763 ± 2018[‡] |
| Dominance, Right, % | 83 |
| Dominant site assessed, % | 42 |

$\bar{X}$ mean, *SD* standard deviation, *BMI* Body Mass Index, *GPAQ* Global Physical Activity Questionnaire.

[†] Missing data for 2 participants.

[‡] Missing data for one participant.

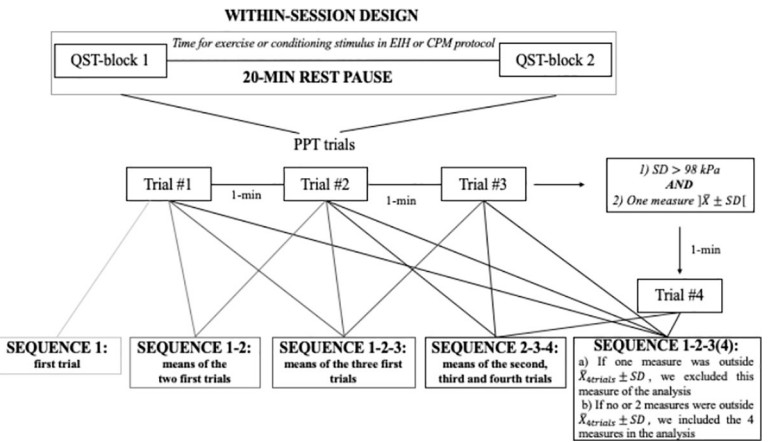

**Fig 1. Study procedure and sequences of measurements analyzed for PPT.**

## 2.3 Quantitative sensory testing (QST)

PPT and TSP measures were collected in the same environment (same room with stable conditions regarding light, temperature and noise) and QST testing order within each testing block was randomized. Side was randomized independently of dominance and measures. If one side (wrist or low back) presented impairments (not related to pain) other than exclusion criteria, the opposite side was tested (1 participant had a limited wrist extension due to a scaphoid fracture 10 years ago [non-painful] and 1 participant had a low back lipoma). QST measures were first tested on the calf or the thigh for familiarization with the procedure.

**2.3.1 Pressure pain threshold.** PPT were assessed with a handheld digital algometer (1-cm$^2$ probe–FPIX, Wagner Instruments, Greenwich, CT, USA) for 22 participants and with a handheld dial algometer (1-cm$^2$ probe–FPK, Wagner Instruments, Greenwich, CT, USA) for 2 participants. Since FPK algometer does not allow to measure between 0 and 1 kg/cm$^2$, FPIX algometer was used for the remaining participants. To determine if the use of FPK algometer affected the reliability, we performed the analysis with and without the first two participants. Considering both analyses provided similar results, we included the first two participants in our analysis. A rate of ~0.5 kg/cm$^2$/s was applied, at two back and two upper limb sites: i) lumbar erector spinae (LES), 2–3 cm laterally to L4/L5 ii) S1 spinous process, iii) dorsal aspect of the wrist on capitatum (WD) and iv) wrist flexors muscles (WF), 10 cm distally from medial humeral epicondyle on a line from medial epicondyle to styloid process of ulna, over the muscles bulk of the wrist flexors. All sites were located and marked before testing. Assessment of the back and the forearm were done in a prone lying position with a pillow under the abdomen and in sitting position with arm supported, respectively. Standardized verbal instructions were given based on German Research Network on Neuropathic Pain (DFNS) recommendations: "This is a test of your sensitivity to deep pain. Now I will press this pressure meter against your back/wrist/forearm and will gradually increase the pressure. Please say ´Now´ as soon as the pressure starts to be painful. Remember that this is not a pain tolerance test, it is a pain threshold test" [32]. Instructions were freely translated into French. PPT were measured three times with one-minute break between measurements. To reduce the variability of the measurement and reduce the impact of potential outlier, a fourth measure was taken if the following two conditions were met: 1) the standard deviation (SD) of the three measures was larger than 1 kg/cm$^2$ and 2) one measure was outside the mean±SD interval (**Fig 1**). The same criteria were applied to the four values to determine if three or four data were kept for analysis. Because of the accuracy limit of the FPIX algometer, PPT data above 11.5 kg/cm$^2$ were considered as 11.5

kg/cm$^2$ (2 trials at S1). For the FPK, PPT between 0 and 1 kg/cm$^2$, were considered as 1 kg/cm$^2$ (4 trials at WD– 2 first participants). This may have caused a small overestimation of the reliability of the PPT measurement at S1 and WD. PPT were transformed from kg/cm$^2$ to kPa (1 kg/cm$^2$ = 98.07 kPa) to facilitate comparisons with the PPT literature.

**2.3.2 Temporal summation of pain.** TSP was tested using a pinprick stimulator (256 mN, MRC Systems GmbH, Heidelberg, Germany) and a series of ten punctuate stimuli at 1 Hz over (i) L4/L5 interspinous process line and (ii) hand dorsum. The 1-Hz frequency was displayed to the rater by a light metronome out of participant's sight. TSP was considered as the difference between the highest numeric rating scale (NRS) (anchored between 0 [no pain] and 10 [worst imaginable pain]) pain through the ten trials and the pain after a single stimulus [27,44]. Standardized verbal instructions were given based on DFNS recommendations: "This is a test of repeated pinpricks. I will now apply a single pinprick. Please give a number between ´0´ and ´10´ for the pain of that stimulus. I will now apply a series of 10 pinpricks in a row. Please give a number between ´0´ and ´10´ for the highest pain over that whole series of 10 pinpricks. This procedure will be repeated 3 times, with a 30-s break between" [32].

## 2.4 Statistical analysis

SPSS software was used for statistical analysis (IBM SPSS 25 for Mac, Armonk, NY, USA). First, Shapiro-Wilk's test and visual appreciation of histogram and Normal Q-Q Plots were used to assess normality of PPT and TSP distributions. Second, presence of outliers was assessed by inspection of a boxplot for values greater than 1.5 box-lengths from the edge of the box (representing a 99.3% confidence interval). Third, homoscedasticity i.e. absence of correlation between the size of error and the magnitude of the observed scores [45,46] was assessed. This was done by visual inspection of Bland-Altman plots of differences between the two values collected at test and retest for each participant by the means of these two values [47,48]. The correlation ($R^2$) between the absolute differences and the means values was also calculated for each variable by a linear regression analysis [49]. In presence of the two following conditions: 1) $R^2$ values greater than 0.1 [49] and 2) $p < 0.05$ for the linear regression model, heteroscedasticity was considered present. Natural logarithmic transformation was applied to data in the presence of non-normality or heteroscedasticity and normality/homoscedasticity were retested to ensure that previous assumptions were met.

**2.4.1 Absolute reliability.** Absolute reliability was assessed using the standard error of the measurement (SEM$_{eas}$) [50]. SEM$_{eas}$ represents the within-subject variation and is defined as "the standard deviation of errors of measurement that is associated with the test scores for a specific group of test takers" [50,51]. SEM$_{eas}$ was estimated as the square root of the mean square error (WMS) calculated by a one-way ANOVA (ANOVA$_{RM}$) applied on test and retest measurements (SEM$_{eas} = \sqrt{WMS}$) [45,52].

To evaluate measures responsiveness, the minimal detectable change for each individual (MDC$_{ind}$) was estimated for PPT and TSP with the following formula: $MDC_{ind} = 1,96 \times SEM_{eas} \times \sqrt{2}$ [53]. MDC for the group (MDC$_{gr}$) was also computed: $MDC_{gr} = MDC_{ind}/\sqrt{n}$, where "n" represents the sample size [54–56]. The values of SEM$_{eas}$, MDC$_{ind}$ and MDC$_{gr}$ were also presented as percentage of pooled means (average of test and retest measurements) [50,52] designated as %SEM$_{eas}$, %MDC$_{ind}$ and %MDC$_{gr}$. %SEM$_{eas}$ and %MDC allow comparisons between studies and facilitate interpretation, since they represent a dimension-less/unit-less measure [50,52]. If SEM$_{eas}$ and MDC were obtained from a log-transformation of the data, an antilog (i.e. exponential ($e^x$)) was applied, resulting in a multiplication/division ($\times/\div$) factor to the test data. This factor indicates the random error component of the mean bias [45]. A constant was added to the distributions before the log-transformation to obtain positive and no-null value.

**2.4.2 Relative reliability.**    The relative reliability was quantified using the intraclass correlation coefficient (ICC). ICC's model 2 was used since each subject was assessed by the same rater and the rater represents the population of possible raters [57]. The ICC was calculated using the Fisher's test of a two-way ANOVA$_{RM}$. "Random effects, absolute agreement, average measures" (2,k) model, for sequences including more than one measurement, and "single-measure" model (2,1) for the sequence of one measurement only were used (see next section for the description of the different sequences of measurements). ICC permits to determine systematic bias between test and retest (difference between means) [46] and ICC 95% confidence interval was calculated to represent ICC variability. An ICC > 0.80 was considered "excellent", between 0.61 and 0.80; "good", between 0.41 and 0.60; "moderate", between 0.21 and 0.40; "acceptable" and 0–0.20 represents "poor" reliability [58].

Considering that the ICC is influenced by the variability of the measurement, the coefficient of variation (CV) of the data was also calculated using the formula: $CV = (SD \div \bar{X}) \times 100$, where SD is the standard deviation of test-retest data and $\bar{X}$ is the mean of test-retest data. CV measures the relative spread of data and helps to consider ICC results in function of its variability [50,52].

**2.4.3 Comparison of different sequences of measurements.**    To determine how many measurements provided the best reliability metrics, five sequences of measurements were considered for PPT (**Fig 1**): first measurement only (**1**), the first two measurements (**1–2**), the first three measurements (**1-2-3**), the first three or the first fourth when applicable (**1-2-3(4)**) and the last two trials including the fourth when applicable (**2-3-4**). The sequence **2-3-4** was included because some studies suggest that the first measure of PPT tends to be higher [36,37,59], but this remains debated [34]. For TSP, three sequences were evaluated: 3 measurements (**1-2-3**), the first two (**1–2**) and the first only (**1**). For all sequences, relative and absolute within-subject short-term test-retest reliabilities were assessed.

# 3. Results

## 3.1 Pain sensitivity outcomes

**3.1.1 Missing data.**    One participant TSP data were excluded of analysis because of a technical problem with the pinprick during the data collection (n = 23). For PPT, twenty-four participants were included in the analysis.

**3.1.2 Assumptions validation and data transformation.**    Two distributions required natural logarithmic (ln) transformation because of non-normality: 1) TSP at the back at retest for sequence 1–2 ($p$ = 0.02) and 2) TSP at the hand at test for sequence 1 ($p$ = 0.03). Data were distributed normally after the transformation. One or two outliers were detected for 20 and 3 out of 52 data distributions, respectively. To determine if outliers affected the reliability analysis, we calculated the reliability with and without outliers. Considering both analyses provided similar reliability that was not affecting the interpretation of the results, outliers were included. All data distributions were homoscedastic.

**3.1.3 Raw PPT and TSP data.**    Group means and SD for PPT and TSP at test and retest are reported in **Table 2**. The results of the one-way ANOVA$_{RM}$ showed that there was no statistically significant difference between test and retest occasions for PPT and TSP at all sites for all sequences (all $p > 0.076$ –**Table 2**).

## 3.2 Reliability analysis

**3.2.1 Absolute reliability for PPT and TSP.**    Considering that (i) MDC$_{gr}$ is proportional to SEM$_{eas}$ and MDC$_{ind}$, and that (ii) %MDC$_{gr}$, %SEM$_{eas}$ and %MDC$_{ind}$ results present the same pattern between sequences, only MDC$_{gr}$ and %MDC$_{gr}$ will be described to facilitate the

**Table 2. Means of PPT and TSP at test and retest occasions for each sequence of measurements.**

| Outcomes | Sites | Sequences of measurements | Test | | Retest | | Test versus retest | |
|---|---|---|---|---|---|---|---|---|
| | | | Mean | SD | Mean | SD | (F) | (p) |
| PPT[a] (kPa) | S1 | 1-2-3(4) | 519.50 | 205.61 | 533.48 | 214.72 | 0.608 | 0.443 |
| | | 1-2-3 | 522.87 | 206.88 | 534.07 | 211.79 | 0.351 | 0.559 |
| | | 2-3-4 | 517.20 | 205.77 | 532.58 | 212.41 | 0.548 | 0.466 |
| | | 1–2 | 521.79 | 216.07 | 530.32 | 210.10 | 0.289 | 0.596 |
| | | 1 | 536.77 | 222.51 | 527.45 | 209.86 | 0.248 | 0.623 |
| | LES | 1-2-3(4) | 547.29 | 199.65 | 551.32 | 169.86 | 0.027 | 0.871 |
| | | 1-2-3 | 551.09 | 201.84 | 557.17 | 172.31 | 0.059 | 0.810 |
| | | 2-3-4 | 549.62 | 207.49 | 559.74 | 175.09 | 0.154 | 0.698 |
| | | 1–2 | 549.52 | 196.11 | 548.79 | 174.74 | 0.001 | 0.976 |
| | | 1 | 553.79 | 199.52 | 553.18 | 181.51 | 0.001 | 0.981 |
| | WD | 1-2-3(4) | 371.67 | 149.28 | 355.95 | 123.96 | 1.072 | 0.311 |
| | | 1-2-3 | 370.54 | 149.03 | 358.91 | 129.38 | 0.631 | 0.435 |
| | | 2-3-4 | 365.02 | 155.98 | 357.37 | 130.20 | 0.225 | 0.640 |
| | | 1–2 | 373.63 | 142.29 | 355.61 | 127.02 | 1.285 | 0.269 |
| | | 1 | 381.89 | 141.27 | 359.97 | 134.01 | 1.821 | 0.190 |
| | WF | 1-2-3(4) | 368.55 | 121.92 | 354.45 | 115.34 | 0.916 | 0.348 |
| | | 1-2-3 | 369.47 | 122.57 | 353.77 | 115.16 | 1.089 | 0.307 |
| | | 2-3-4 | 361.41 | 127.11 | 349.83 | 115.80 | 0.509 | 0.483 |
| | | 1–2 | 371.86 | 126.16 | 356.19 | 115.64 | 0.969 | 0.335 |
| | | 1 | 389.94 | 135.22 | 365.73 | 127.69 | 1.518 | 0.230 |
| TSP[b] (NRS) | Hand | 1-2-3 | 2.29 | 1.45 | 2.61 | 1.51 | 3.461 | 0.076 |
| | | 1–2 | 2.21 | 1.43 | 2.47 | 1.52 | 1.731 | 0.202 |
| | | 1 | 2.24** | 1.63** | 2.28** | 1.71** | 0.011* | 0.917* |
| | Back | 1-2-3 | 1.92 | 1.32 | 1.95 | 1.58 | 0.015 | 0.905 |
| | | 1–2 | 1.85** | 1.31** | 1.98** | 1.69** | 0.016* | 0.902* |
| | | 1 | 1.96 | 1.32 | 1.91 | 1.71 | 0.022 | 0.884 |

*PPT* pressure pain threshold, *TSP* temporal summation of pain, *NRS* numeric rating scale, *SD* standard deviation, *S1* spinous process of S1, *LES* lumbar erector spinae, *WD* dorsal aspect of the wrist, *WF* wrist flexors muscles.

* ANOVA were done on natural log-transformation of the data distributions.

** Means and SD of the non-transformed raw data are presented to facilitate interpretation.

F Fisher's test from a one-way ANOVA$_{RM}$.

p one-way ANOVA$_{RM.}$

[a] n = 24.

[b] n = 23.

presentation of the results. Each metric is detailed in Table 3. In general, differences in the measures of absolute reliability for PPT are small across the sequences and seem to be dependent on the site tested (Fig 2). For S1, MDC$_{gr}$ varied from 31.10 kPa for sequence 1–2 to 40.71 kPa for sequence 2-3-4 (%MDC$_{gr}$: 5.91% (1–2) to 7.76% (2-3-4)). For LES, MDC$_{gr}$ ranged from 46.13 kPa (1–2) to 50.56 kPa (1) (%MDC$_{gr}$: 8.40% (1–2) to 9.14% (1)). For WD, MDC$_{gr}$ varied from 28.71 kPa (1-2-3) to 31.84 kPa (1) (%MDC$_{gr}$: 7.87% (1-2-3) to 8.58% (2-3-4)). For WF, MDC$_{gr}$ ranged from 28.87 kPa (1-2-3(4)) to 38.52 kPa (1) (%MDC$_{gr}$: 7.99% (1-2-3(4)) to 10.19% (1)). Sequences 1-2-3(4) and 1-2-3 presented similar results at WD and WF and seem to present a smaller measurement error compared to other sequences regarding absolute reliability. At the low back (S1 and LES), sequence **1–2** presented the lowest MDC$_{gr}$ compared to

**Table 3. Within-session reliability parameters for PPT for each site using different sequences of measurements (n = 24).**

| Sites PPT | Sequences | ICC (95%CI) | CV (%) | $SEM_{eas}$ | | $MDC_{ind}$ | | $MDC_{gr}$ | |
|---|---|---|---|---|---|---|---|---|---|
| | | | | kPa | % | kPa | % | kPa | % |
| S1 | 1-2-3(4) | 0.96 (0.90–0.98) | 39.52 | 62.08 | 11.79 | 172.08 | 32.68 | 35.12 | 6.67 |
| | 1-2-3 | 0.95 (0.89–0.98) | 39.21 | 65.49 | 12.39 | 181.54 | 34.35 | 37.06 | 7.01 |
| | 2-3-4 | 0.94 (0.86–0.97) | 39.44 | 71.95 | 13.71 | 199.45 | 38.00 | 40.71 | 7.76 |
| | 1–2 | 0.97 (0.92–0.99) | 40.09 | 54.97 | 10.45 | 152.37 | 28.96 | 31.10 | 5.91 |
| | 1 | 0.91 (0.81–0.96) | 40.22 | 64.83 | 12.18 | 179.90 | 33.77 | 36.68 | 6.89 |
| LES | 1-2-3(4) | 0.89 (0.74–0.95) | 33.39 | 85.21 | 15.51 | 236.20 | 43.00 | 48.21 | 8.78 |
| | 1-2-3 | 0.89 (0.73–0.95) | 33.51 | 86.56 | 15.62 | 239.93 | 43.30 | 48.98 | 8.84 |
| | 2-3-4 | 0.88 (0.73–0.95) | 34.25 | 89.36 | 16.11 | 247.70 | 44.66 | 50.56 | 9.12 |
| | 1–2 | 0.90 (0.76–0.96) | 33.46 | 81.53 | 14.85 | 225.98 | 41.15 | 46.13 | 8.40 |
| | 1 | 0.79 (0.57–0.90) | 34.09 | 89.37 | 16.15 | 247.71 | 44.76 | 50.56 | 9.14 |
| WD | 1-2-3(4) | 0.92 (0.82–0.97) | 37.38 | 52.62 | 14.46 | 145.85 | 40.09 | 29.77 | 8.18 |
| | 1-2-3 | 0.93 (0.84–0.97) | 37.89 | 50.74 | 13.91 | 140.66 | 38.57 | 28.71 | 7.87 |
| | 2-3-4 | 0.92 (0.82–0.97) | 39.37 | 55.92 | 15.48 | 154.99 | 42.91 | 31.64 | 8.76 |
| | 1–2 | 0.91 (0.79–0.96) | 36.68 | 55.07 | 15.10 | 152.66 | 41.87 | 31.16 | 8.55 |
| | 1 | 0.83 (0.65–0.92) | 36.84 | 56.28 | 15.17 | 155.99 | 42.05 | 31.84 | 8.58 |
| WF | 1-2-3(4) | 0.90 (0.77–0.96) | 32.54 | 51.02 | 14.11 | 141.42 | 39.12 | 28.87 | 7.99 |
| | 1-2-3 | 0.89 (0.76–0.95) | 32.61 | 52.12 | 14.41 | 144.47 | 39.95 | 29.49 | 8.15 |
| | 2-3-4 | 0.88 (0.73–0.95) | 33.86 | 56.22 | 15.81 | 155.84 | 43.82 | 31.81 | 8.95 |
| | 1–2 | 0.88 (0.74–0.95) | 32.96 | 55.14 | 15.15 | 152.84 | 41.99 | 31.20 | 8.57 |
| | 1 | 0.73 (0.47–0.87) | 34.59 | 68.08 | 18.02 | 188.71 | 49.94 | 38.52 | 10.19 |

*ICC* intraclass correlation coefficient, with a 95% confidence interval, *CV* coefficient of variation, $SEM_{eas}$ standard error of the measurement, $MDC_{ind}$ individual minimal detectable change, $MDC_{gr}$ group minimal detectable change, *S1* spinous process of S1, *LES* lumbar erector spinae, *WD* dorsal aspect of the wrist, *WF* wrist flexors muscles.

other sequences although the difference remains minimal. In terms of the different sites, S1 presented the lowest measurement error compared to the three other sites that were very similar. Again, these differences were small.

For TSP, absolute reliability is depicted in **Table 4**. $MDC_{gr}$ at the hand varied from 0.33 for sequence 1-2-3 to 0.40/10 for sequence 1–2 (%$MDC_{gr}$: 13.58% (1-2-3) to 17.29% (1–2)). At the back, $MDC_{gr}$ ranged from 0.46 for sequence 1-2-3 to 0.57/10 for sequence **1** (%$MDC_{gr}$: 23.52% (1-2-3) to 29.22% (1)). An increase in the number of measurements for TSP leads to an increased absolute reliability. %$MDC_{gr}$ was lower at the hand compared to the back. However, although we presented the non-transformed data in **Table 4**, comparisons with other sequences must be done with caution. For log-transformation data, calculation of %$SEM_{eas}$ and %MDC was not applicable because transformed results do not correspond to the same ratio scale [54,55]. Antilog factor (×/÷) derived from $SEM_{eas}$ and $MDC_{ind}$ were also calculated and presented in **Table 4**.

**3.2.2 Relative reliability for PPT and TSP.** For PPT, almost all ICC values are above 0.80, between 0.80 and 0.97, denoting that almost all sequences presented an excellent relative reliability, except sequence 1 for PPT at LES and WF, for which an acceptable reliability was observed with ICC of 0.79 and 0.73 respectively (**Table 3**). Sequence 1 presented lower ICCs for all sites, associated to large confidence intervals at LES, WD and WF. As illustrated in **Fig 3**, ICCs of sequence 1–2 are larger at S1 and LES than others. ICCs for WD and WF were similar among sequences, except for sequence 1 that presents with lower ICCs.

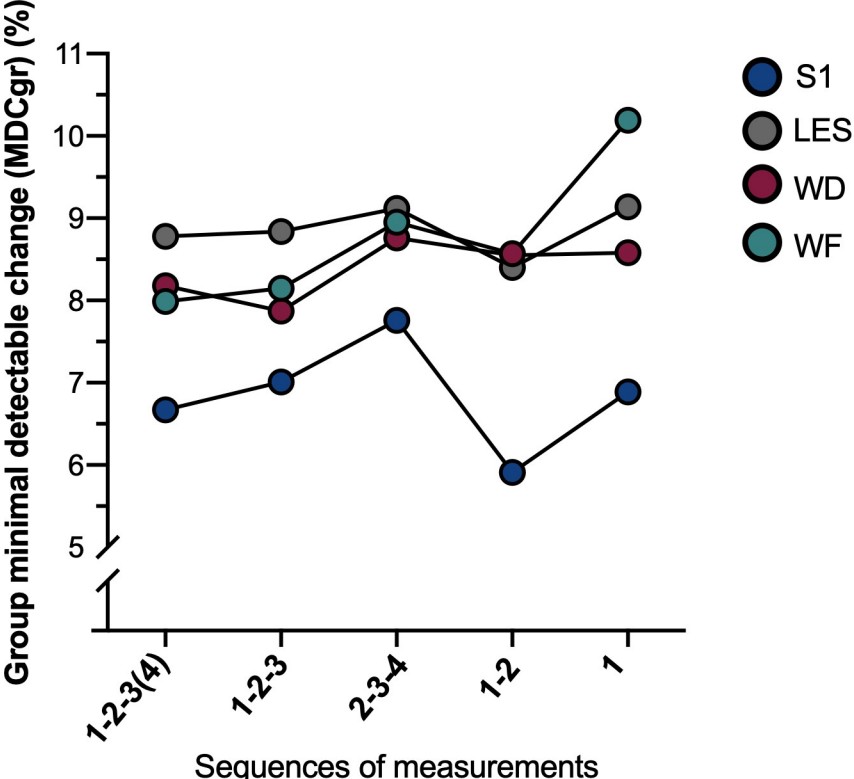

**Fig 2. %MDC$_{gr}$ for PPT at all sites for the five sequences of measurements tested.**

For TSP, results are depicted in **Table 4**. Sequences 1–2 and 1-2-3 showed an excellent reliability with ICC ranged between 0.81 and 0.91 and narrow confidence intervals. Visually, larger ICC confidence intervals were present at the back compared to the hand. As illustrated

**Table 4. Within-session reliability parameters for TSP for each site using different sequences of measurements (n = 23).**

| Sites TSP | Sequences of measurements | ICC (95%CI) | CV (%) | SEM$_{eas}$ | | MDIC$_{ind}$ | | MDIC$_{gr}$ | |
|---|---|---|---|---|---|---|---|---|---|
| | | | | NRS | % | NRS | % | NRS | % |
| **Hand** | 1-2-3 | 0.91 (0.79–0.96) | 60.29 | 0.58 | 23.50 | 1.59 | 65.16 | 0.33 | 13.58 |
| | 1–2 | 0.88 (0.71–0.95) | 62.74 | 0.70 | 29.91 | 1.94 | 82.91 | 0.40 | 17.29 |
| | 1* (non-transformed) | 0.73 (0.46–0.88) | 73.09 | 0.86 | 38.03 | 2.38 | 105.39 | 0.50 | 21.98 |
| | 1 (log$_n$) | 0.78 (0.54–0.90) | NA | 0.15 | NA | 0.41 | NA | 0.09 | NA |
| | 1 (antilog) | | | ×/÷1.16 | | ×/÷1.51 | | | |
| **Back** | 1-2-3 | 0.83 (0.59–0.93) | 74.47 | 0.79 | 40.70 | 2.18 | 112.81 | 0.46 | 23.52 |
| | 1-2* (non-transformed) | 0.81 (0.55–0.92) | 78.32 | 0.85 | 44.60 | 2.37 | 123.63 | 0.49 | 25.78 |
| | 1–2 (log$_n$) | 0.81 (0.55–0.92) | NA | 0.17 | NA | 0.47 | NA | 0.10 | NA |
| | 1–2 (antilog) | | | ×/÷1.18 | | ×/÷1.60 | | | |
| | 1 | 0.58 (0.23–0.80) | 78.12 | 0.98 | 50.56% | 2.71 | 140.15 | 0.57 | 29.22 |

*ICC* intraclass correlation coefficient, with a 95% confidence interval, *CV* coefficient of variation, *SEM$_{eas}$* standard error of the measurement, *MDIC$_{ind}$* individual minimal detectable change, *MDIC$_{gr}$* group minimal detectable change, *NA*: not applicable.

* Reliability outcomes are presented for non-transformed data distributions in order to facilitate interpretation but considerations of these data must be done cautiously because the assumption of the normality of the distribution was violated.

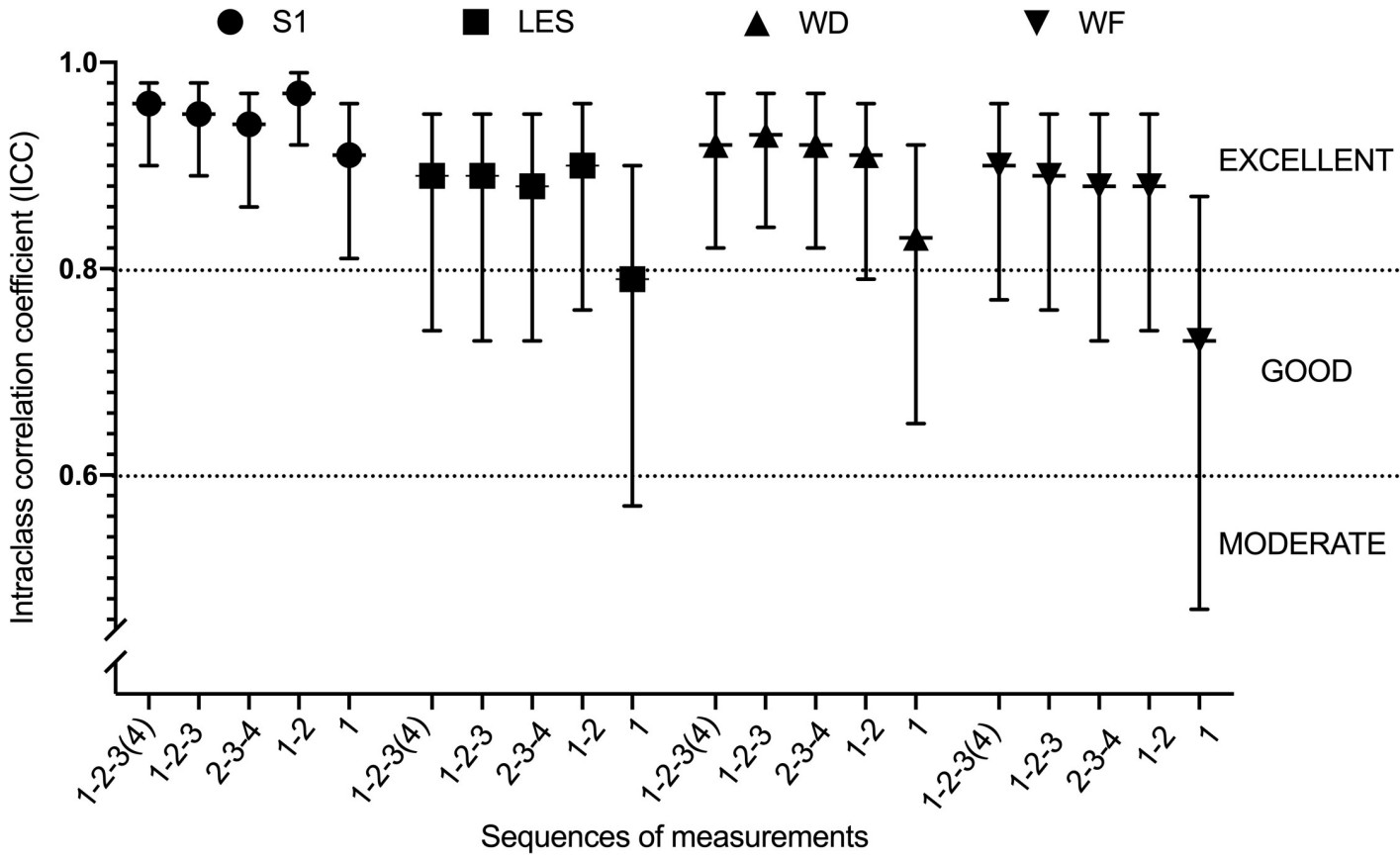

**Fig 3. ICC for PPT for all sites by each sequence of measurements.**

in **Fig 4**, there was a large difference between sequence 1 and 1-2-3 for ICC at the two sites, with sequence 1-2-3 presenting larger ICCs compared to sequence 1.

## 4. Discussion

The objectives of this study were to determine the reliability of PPT and TSP and to determine the sequence of measurements providing the best reliability metrics. The design of the study was established to measure the minimal change exceeding the variability of PPT and TSP technique within a session (20 minutes apart). Our results could be used in future pain research to confirm the presence of 'true' hypo- or hyperalgesia (i.e. change over normal variability) for paradigms such as CPM or EIH. In addition, we established that two measures of PPT and three measures of TSP reduced the measurement error and demonstrated an excellent relative reliability.

### 4.1 PPT reliability

Four studies investigated the intra-rater reliability of PPT at the low back and at the wrist in healthy participants [34–37], but only two of them [34,37] evaluated the within-session test-retest reliability of PPT at the low back. The two other studies based their reliability analysis on the comparison of two or three consecutive measurements from a single block of measurements. Therefore, their results are not suitable in a context of pain modulation evaluation such as CPM/EIH and cannot be directly compared to our results. Balaguier et al. [34] evaluated

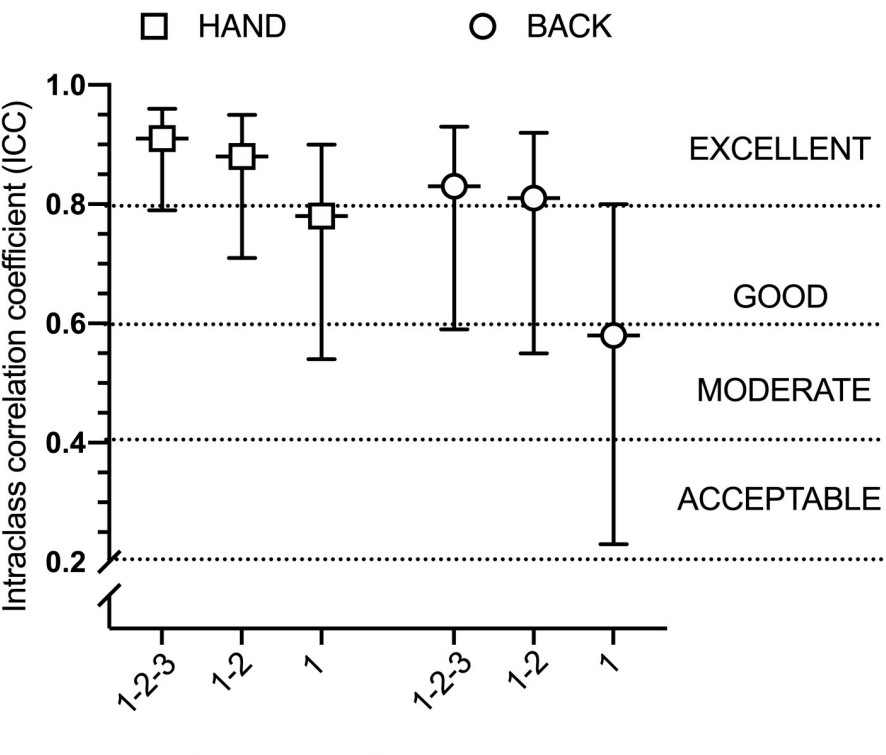

**Fig 4. ICC for TSP at the hand and at the back by each sequence of measurements.**

absolute reliability at the low back over two sessions separated by one hour and observed a MDC$_{ind}$ ranging from 94 to 253 kPa. These results are similar to our findings for the low back that varied from 152.37 to 247.71 kPa. In addition, our ICCs for PPT are consistent with the literature. One study reported excellent relative reliability (ICCs: 0.86 to 0.99) at 14 anatomical locations at the low back and another study reported good to excellent reliability (ICC: 0.40 to 0.99) for three sites over lumbar *erector spinae* (L1, L3, L5) [37]. Previous results must be interpreted with caution because of the limited sample size (n = 5 [37] and n = 15 [34]).

### 4.2 TSP reliability

Only three studies investigated mechanical TSP reliability [60–62] but none assessed the low back area. In two studies [60,62], 5 consecutive series of ten punctuate stimuli were used instead of two blocks of 3 series tested 20-min apart. One study [60] observed a poor reliability for mechanical TSP at the face, hand and foot and another study reported poor to good intra-rater reliability at the tongue, face and gingiva [62]. The third study investigated the test-retest reliability of mechanical TSP at the hand within a two-weeks periods and observed an acceptable reliability in younger adults and a moderate to good reliability in older adults [61]. These findings are inconsistent with the excellent reliability observed for mechanical TSP at back and hand sites in our study. This discrepancy can be caused by differences in sites tested, study designs and calculation of TSP. In two of the above-mentioned studies [60,62], TSP reliability was analyzed using the wind-up ratio calculation (WUR = the mean pain ratings of trains divided by the mean pain rating to single stimulus). WUR cannot be calculated if the single pinprick stimulus is rated as non-painful (NRS = 0/10, meaning a null denominator), leading to an undefined division and limiting the number of patients included in the analysis (e.g. up

to ~27% participants excluded of the analysis [60]). We considered that the subtraction method used for TSP also reflect the facilitation process of nociceptive inputs but with the advantage to include more participants into the analysis.

Antilog factors were calculated for two TSP sequences considering that a log-transformation was applied in presence of normality violation. For example, an antilog factor for the $MDC_{ind}$ of $\times/\div 1.60$ for method 1–2 at the back was obtained. This factor is applied on the test score to obtain lower/higher limits representing the level below/above which the retest has to change to be considered as a real change. For example, if a TSP of 2/10 is measured at test, it implies that a TSP $\geq 3.20/10$ ($2\times1.60$) or TSP $\leq 1.25/10$ ($2\div1.60$) at retest for the same participant are considered as a true change [45,47,54]. Some methodological limitations derived from the application of the antilog factor, for example, the antilog factor could not be used if TSP at test was rated 0/10. Also, considering that TSP represents ordinal data, interpretation appears to be less precise. Therefore, antilog factor calculation constitutes an alternative way to analyze reliability in presence of normality violation, but its concrete and clinical applicability remains challenging with TSP data.

### 4.3 Comparison of sequences of measurements

For PPT, there was substantial heterogeneity in terms of the sequences of measurements producing the least measurement error, which differs in function of the site tested. Absolute reliability with sequence 1–2 was superior for S1 and LES, sequence 1-2-3 for WD and 1-2-3-(4) for WF. However, these differences were minimal across sequences. In addition, all the sequences tested for PPT demonstrated an excellent relative reliability at all sites, except sequence 1 which showed a good relative reliability at LES and WF. Our results are generally in accordance with current literature suggesting that the mean of two or three consecutive measurements are enough to provide reliable PPT [34,36]. However, one study proposed that using only the first measure constitutes an excellent method [34] in opposition to our findings. Some studies observed that the first PPT measurement tends to be significantly higher compared to the subsequent ones and recommended to exclude it from the analysis to reduce the variability [36,37,59]. Our results suggest that excluding the first measurement (2-3-4) did not reduce the measurement error. However, when the first measure was used alone, usually, reliability worsen, meaning that adding only one more measurement seemed sufficient to stabilize the measure. Disparity regarding the first measure can be caused by the fact that some studies did not include familiarization trial prior to PPT evaluation [36,37] or evaluated PPT at different sites than those evaluated in the present study (i.e. *biceps brachii*) [59]. Thus, we recommend to measure PPT at least twice in each block of testing to improve reliability.

For TSP, our results suggest that an increase in the number of measurements leads to an increased absolute reliability. Two and three measurements showed an excellent relative reliability at the hand and the back, in contrast a single measurement demonstrated a good relative reliability at the hand and moderate at the low back. These results may have been influenced by the sample heterogeneity, as represented by high coefficients of variation for TSP. Although some research groups recommend to measure 5 series of ten punctuate stimuli [32,60,62], our results suggest that a sequence of 3 series separated by 30 seconds provides an excellent reliability. Taking 3 measures instead of 5 may reduce the duration of the test and minimize skin irritation due to repeated pinprick stimulations. Therefore, we suggest to take at least three TSP measurement to improve the TSP reliability.

### 4.4 Applicability of absolute reliability in research

The MDC obtained in our study provides specific threshold to determine if change in PPT or TSP in a within-session design exceeds the variability of these measures for a group ($MDC_{gr}$)

or an individual ($MDC_{ind}$). In research, MDC can serve as a cut-off value to determine if the pain sensitivity change (as measured by PPT or TSP) exceeds measurement error and can be considered as true hypo- or hyperalgesia following the conditioning stimulus in pain modulation paradigm (e.g. CPM and EIH). For instance, a previous study investigating EIH reported a significant within-session PPT change of 29.78 kPa at the low back in healthy controls following a repetitive lifting task [11]. We determined that the lowest $MDC_{gr}$ at the low back was 46.13 kPa. Thus, a change inferior to this value cannot be considered as a real change despite statistical significance.

Also, the response to conditioning stimulus (e.g. CPM) has been used to stratify healthy and chronic pain participants in function of the change in pain sensitivity (decrease vs. increase) [9,63,64] suggesting a bias toward inhibitory or facilitatory descending control (e.g. anti- vs pro-nociceptive). However, this stratification is usually done without considering the measurement error. This may result in stratifying participants for whom changes remain within the measurement error (i.e. pain sensitivity did not change following conditioned stimulus). Our $MDC_{ind}$ (or $\%MDC_{ind}$) could be used in future studies using a similar design (CPM/EIH) as cut-off values to subgroup participants. For example, an individual change of ~ 35% is necessary to exceed the $\%MDC_{ind}$ at S1. Considering that this change is large, it questions the validity of stratification methods using PPT.

## 4.5 Methodological considerations

Our study did not conduct the reliability analysis in different groups for each sequence and this could have underestimated PPT and TSP measurement variability. Also, considering that this study was conducted in a healthy pain-free population, current results are not generalizable to other populations. It is acknowledged that factors such as race/ethnicity, sex (e.g. phases of menstrual cycle [65]) and age could influence pain sensitivity [66,67], but considering that we measured reliability of pain sensitivity in a single session rather than pain sensitivity *per se*, the effect of these factors on our results remain limited. Future studies must be conducted in chronic pain participants, such as chronic low back pain.

## 4.6 Summary

This study observes that PPT and TSP at back and hand sites have small error of measurement and an excellent relative reliability using a within-session test-retest design. Our results also suggest that at least two PPT and three TSP consecutive measures are needed to optimize reliability and these recommendations may be used in future research and in clinical practice. Our results also provide cut-off values that may be used with pain modulation paradigms such as CPM and EIH to confirm that changes following conditioning stimulus exceed PPT and TSP measurement error (true hypo-/hyperalgesia). Further studies are warranted to investigate the within-session test-retest reliability of these parameters in chronic pain populations.

## Author Contributions

**Conceptualization:** Catherine Mailloux, Louis-David Beaulieu, Timothy H. Wideman, Hugo Massé-Alarie.

**Data curation:** Catherine Mailloux.

**Formal analysis:** Catherine Mailloux.

**Funding acquisition:** Hugo Massé-Alarie.

**Investigation:** Catherine Mailloux.

**Methodology:** Catherine Mailloux, Louis-David Beaulieu, Timothy H. Wideman, Hugo Massé-Alarie.

**Project administration:** Catherine Mailloux, Hugo Massé-Alarie.

**Resources:** Catherine Mailloux, Hugo Massé-Alarie.

**Software:** Hugo Massé-Alarie.

**Supervision:** Louis-David Beaulieu, Timothy H. Wideman, Hugo Massé-Alarie.

**Validation:** Hugo Massé-Alarie.

**Visualization:** Hugo Massé-Alarie.

**Writing – original draft:** Catherine Mailloux, Hugo Massé-Alarie.

**Writing – review & editing:** Louis-David Beaulieu, Timothy H. Wideman, Hugo Massé-Alarie.

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
