## [Decision Letter · Decision Letter 0]

3 Dec 2020

PONE-D-20-27453

Within-session test-retest reliability of pressure pain threshold and mechanical temporal summation in healthy subjects

PLOS ONE

Dear Dr. Massé-Alarie,

Thank you for submitting your manuscript to PLOS ONE. After careful consideration, we feel that it has merit but does not fully meet PLOS ONE’s publication criteria as it currently stands. Therefore, we invite you to submit a revised version of the manuscript that addresses the points raised during the review process.

We look forward to receiving your revised manuscript.

Kind regards,

Alison Rushton

Academic Editor

PLOS ONE

Additional Editor Comments:

Please address the reviewers' comments detailed below.

Reviewers' comments:

Reviewer's Responses to Questions

**Comments to the Author**

1. Is the manuscript technically sound, and do the data support the conclusions?

Reviewer #1: Yes

Reviewer #2: Yes

Reviewer #3: Yes

2. Has the statistical analysis been performed appropriately and rigorously? 

Reviewer #1: Yes

Reviewer #2: Yes

Reviewer #3: Yes

3. Have the authors made all data underlying the findings in their manuscript fully available?

Reviewer #1: Yes

Reviewer #2: Yes

Reviewer #3: Yes

4. Is the manuscript presented in an intelligible fashion and written in standard English?

Reviewer #1: Yes

Reviewer #2: Yes

Reviewer #3: Yes

5. Review Comments to the Author

Reviewer #1: 

Comments:

- The authors propose a very interesting study to determine how many measurements provide the best reliability metrics. Thus, the authors presented a well-designed study design, in which they evaluated the minimum change that exceeds the variability of the pressure pain threshold (PPT) technique and the mechanical time sum of pain (TSP) with 3 measures from two tested blocks with a 20-minute break and retest.

- The authors used convenience sampling. Even if it's only for the studied group, the main consequence is to generalize the study. Although it can offer us valuable information in countless circumstances, especially when there are no fundamental reasons that differentiate accessible individuals. The type of sampling for convenience in this study I believe will not introduce bias in relation to the total population, the results obtained can be used for the studied universe (pain in specific sites).

Considerations and one question:

In the item materials and methods, in the description of the gender of the participants (line 92), there is an incoherence of the authors in relation to table 1 (descriptive statistics - line 106). The authors reported 12 females from the 24 participants, in table 1 they report ills (50%). Exclusion criteria are presented for females. Was the study carried out with both? It is not clear. Table 1 must be offset.

-If the study was conducted with females, did they not exclude menstrual periods and menopause? These conditions create pain. Could they interfere with the measurements?

The age range in the age of 18 to 65 years, could generate bias, although it was 28.3 the average age.

Reviewer #2: 

Please provide operational definition for healthy subjects. Clarify the inclusion criteria....the cause/s of pain in the subjects included in the study. Please add discussion about patient characteristics, any influencing factors that could have affected the study result

Reviewer #3: 

The study was purported to determine the absolute and relative intra-rater within-session test-retest reliability of pressure pain threshold (PPT) and temporal summation of pain (TSP) and the influence of the sequence of the measurements. Authors established that at least two PPT and three TSP consecutive measures are needed to optimize reliability. The study was well conducted, well written to clearly communicate the results and has added additional evidence to the reliability of pain measurements at the low back and forearm in a healthy population. However, there are a few suggestions to consider to improve the manuscript.

Results

Line 249 – 251: The second sentence seems incomplete. Are the first-two sentences supposed to be linked as one sentence?

Figure 2. The color coding for LES and WF looks identical and difficult to follow. Consider using a different shape(s).

Discussion

Line 344: The sentence “Some methodological limitations derived from the application of the antilog factor” looks incomplete.

Lines 359 – 364. It’s interesting that there are some conflicting outcomes with the first PPT measurement from other studies. It would be great to briefly discuss possible factors in these studies that could have led to the different findings relative to the finding in your study.

6. PLOS authors have the option to publish the peer review history of their article (what does this mean?). If published, this will include your full peer review and any attached files.

Reviewer #1: **Yes: **Izabel Cristina Custódio de Souza

Reviewer #2: No

Reviewer #3: No

---

## [Author Response · Author response to Decision Letter 0]

22 Dec 2020

Ms. Ref. No.: PONE-D-20-27453

Title: Within-session test-retest reliability of pressure pain threshold and mechanical temporal summation in healthy subjects

PLOS One

Reviewers' comments:

We thank the Reviewers for their comments that will substantially improve the quality of the manuscript. We address attentively each one of them. Our responses and modifications are presented below. Changes in the manuscript are red-typed.

Reviewer #1: 

Comments:

- The authors propose a very interesting study to determine how many measurements provide the best reliability metrics. Thus, the authors presented a well-designed study design, in which they evaluated the minimum change that exceeds the variability of the pressure pain threshold (PPT) technique and the mechanical time sum of pain (TSP) with 3 measures from two tested blocks with a 20-minute break and retest.

- The authors used convenience sampling. Even if it's only for the studied group, the main consequence is to generalize the study. Although it can offer us valuable information in countless circumstances, especially when there are no fundamental reasons that differentiate accessible individuals. The type of sampling for convenience in this study I believe will not introduce bias in relation to the total population, the results obtained can be used for the studied universe (pain in specific sites).

Considerations and one question:

In the item materials and methods, in the description of the gender of the participants (line 92), there is an incoherence of the authors in relation to table 1 (descriptive statistics - line 106). The authors reported 12 females from the 24 participants, in table 1 they report ills (50%). Exclusion criteria are presented for females. Was the study carried out with both? It is not clear. Table 1 must be offset.

RESPONSE: At the line 92, the number of females is presented to specify the sex of the recruited participants. This study was therefore carried in 12 females and 12 males, for a total recruited sample of 24 participants. The exclusion criteria presented after are for both (females and males). To reduce confusion, Table 1 was modified to state the number of both males and females. 

-If the study was conducted with females, did they not exclude menstrual periods and menopause? These conditions create pain. Could they interfere with the measurements?

The age range in the age of 18 to 65 years, could generate bias, although it was 28.3 the average age.

RESPONSE: We agree with Reviewer #1 that phases of menstrual cycle and menopause could create pain in females and could influence pain sensitivity. Indeed, a study of Pogatzki-Zhan et al. showed that pinprick pain is greater in the luteal phase of the menstrual cycle and predicted by progesterone (1). We did not determine menstrual periods and menopause as exclusion criteria in females because pain related to menstrual periods is mostly short-time and episodic, considered as intermittent pain rather than chronic pain. However, we agree that it may introduce variability in the results. We excluded participants presented pain located anywhere in the body lasting more than three months. This substantially reduced the possibility of central adaptations (within the central nervous system) related to pain which ensure that our pain-free sample is homogenous and that pain-free participants are really healthy. Also, considering we studied reliability, and that test and retest sessions were performed in the same day, we believe that impact of phases of menstrual cycle on results is limited. We still added a sentence in the Methodological section to address these points:

p. 19-20; ln 403-407: “It is acknowledged that factors such as race/ethnicity, sex (e.g. phases of menstrual cycle (1)) and age could influence pain sensitivity (2, 3), but considering that we measured reliability of pain sensitivity in a single session rather than pain sensitivity per se, the effect of these factors on our results remain limited.”

Reviewer #2: 

Please provide operational definition for healthy subjects. Clarify the inclusion criteria....the cause/s of pain in the subjects included in the study. Please add discussion about patient characteristics, any influencing factors that could have affected the study result

RESPONSE: We understand the concern of the Reviewer #2. Participants included in our study did not report any clinical pain, they were pain-free participants. Pain was induced using quantitative sensory testing (QST). We added all the other participants characteristics collected in Table 1 in order to better describe participants (body mass index (BMI) and score to Global Physical Activity Questionnaire (GPAQ)). We also added a sentence in Material and Methods to introduce these characteristics. 

Ln 105-107 p.5: ‘’The body mass index was calculated for each participant and the Global Physical Activity Questionnaire (GPAQ) was self-administered to rate the level of physical activity (4)’’.

We added another sentence in the Discussion to specify that pain sensitivity could be affected by personal characteristics and factors, but considering the within-subject design used, we estimate that their influence on results is limited. 

Ln 403-407 p.19-20: “It is acknowledged that factors such as race/ethnicity, sex (e.g. phases of menstrual cycle (1)) and age could influence pain sensitivity (2, 3), but considering that we measured reliability of pain sensitivity in a single session rather than pain sensitivity per se, the effect of these factors on our results remain limited.”

Reviewer #3: 

The study was purported to determine the absolute and relative intra-rater within-session test-retest reliability of pressure pain threshold (PPT) and temporal summation of pain (TSP) and the influence of the sequence of the measurements. Authors established that at least two PPT and three TSP consecutive measures are needed to optimize reliability. The study was well conducted, well written to clearly communicate the results and has added additional evidence to the reliability of pain measurements at the low back and forearm in a healthy population. However, there are a few suggestions to consider to improve the manuscript.

Results

Line 249 – 251: The second sentence seems incomplete. Are the first-two sentences supposed to be linked as one sentence?

RESPONSE: We totally agree with the Reviewer #3. The sentence was reformulated as follows.

Ln 253-255, p.12: “Considering that (i) MDCgr is proportional to SEMeas and MDCind, and that (ii) %MDCgr, %SEMeas and %MDCind results present the same pattern between sequences, only MDCgr and %MDCgr will be described to facilitate the presentation of the results”. 

Figure 2. The color coding for LES and WF looks identical and difficult to follow. Consider using a different shape(s).

RESPONSE: As recommended by Reviewer #3, the colors of the Figure 2 was modified to better distinguish each studied site. Same shapes was kept to ensure the uniformity of the figure.

Discussion

Line 344: The sentence “Some methodological limitations derived from the application of the antilog factor” looks incomplete.

RESPONSE: As recommended by the Reviewer #3, the sentence was modified as follows.

Ln 348-350, p.17: “Some methodological limitations derived from the application of the antilog factor, for example, the antilog factor could not be used if TSP at test was rated 0/10’’.

Lines 359 – 364. It’s interesting that there are some conflicting outcomes with the first PPT measurement from other studies. It would be great to briefly discuss possible factors in these studies that could have led to the different findings relative to the finding in your study.

RESPONSE: As recommended, the next sentence was added to suggest an explanation to this disparity between our results and the current literature. 

Ln 367-370, p.18: “Disparity regarding the first measure can be caused by the fact that some studies did not include familiarization trial prior to PPT evaluation (5, 6) or evaluated PPT at different sites than those evaluated in the present study (i.e. biceps brachii) (7).”

---

## [Editor Report · Decision Letter 1]

28 Dec 2020

Within-session test-retest reliability of pressure pain threshold and mechanical temporal summation in healthy subjects

PONE-D-20-27453R1

Dear Dr. Massé-Alarie,

We’re pleased to inform you that your manuscript has been judged scientifically suitable for publication and will be formally accepted for publication once it meets all outstanding technical requirements.

Kind regards,

Alison Rushton

Academic Editor

PLOS ONE

Additional Editor Comments (optional):

Thank you for addressing all comments from the reviewers to a satisfactory level.

I hope that you agree the changes have improved the quality of the manuscript further.

Reviewers' comments:

No further review required.

---

## [Editor Report · Acceptance letter]

4 Jan 2021

PONE-D-20-27453R1 

Within-session test-retest reliability of pressure pain threshold and mechanical temporal summation in healthy subjects 

Dear Dr. Massé-Alarie:

I'm pleased to inform you that your manuscript has been deemed suitable for publication in PLOS ONE. Congratulations! Your manuscript is now with our production department. 

Kind regards, 

on behalf of

Professor Alison Rushton 

Academic Editor

PLOS ONE